# Impact of River-Reservoir Hybrid System on Zooplankton Community and River Connectivity

Eui-Jeong Ko [1,2], Eunsong Jung [1], Yuno Do [3], Gea-Jae Joo [1], Hyun-Woo Kim [4] and Hyunbin Jo [1,5,*]

1. Department of Integrated Biological Science, Pusan National University, Busan 46241, Korea; nangyj@nie.re.kr (E.-J.K.); vjvmf010@gmail.com (E.J.); gjjoo@pusan.ac.kr (G.-J.J.)
2. National Institute of Ecology, Seo-Cheon 33657, Korea
3. Department of Biological Sciences, Kongju National University, Gongju 32588, Korea; doy@kongju.ac.kr
4. Department of Environmental Education, Sunchon National University, Suncheon 57922, Korea; hwkim@sunchon.ac.kr
5. Institute for Environment and Energy, Pusan National University, Busan 46241, Korea
* Correspondence: prozeva@pusan.ac.kr

**Abstract:** Anthropogenic connectivity regulation in rivers, such as via weirs and dams, affects the plankton community. We hypothesized that the longitudinal similarity of the zooplankton community in a river could change in a river–reservoir hybrid system (RRHS). The impact of weir construction on zooplankton communities in terms of species diversity, abundance, and community structure was examined biweekly at six sites on the Nakdong River for 14 years (before construction: 2002–2008; after construction: 2012–2018). We checked time-series alignment using a dynamic time-warping method between longitudinal survey sites. After RRHS, the zooplankton community showed an increasing number of species. However, RRHS decreased the longitudinal similarity in terms of number of zooplankton species and population density. Our results demonstrate the negative effect of lateral infrastructures on zooplankton populations due to river fragmentation and habitat alteration.

**Keywords:** dynamic time warping; longitudinal connectivity; weir effect; long-term monitoring; Nakdong River

## 1. Introduction

Globally, rivers are being transformed from free-flowing reaches into continuous reservoir clusters as lateral barriers are constructed to meet the demand for increasing water consumption. These clusters are known as river–reservoir hybrid systems (RRHSs) [1]. These changes are part of the efforts to sustain a stable water supply and aquatic ecosystem for growing populations [2]. However, lateral structures have been found to affect the limnology and hydrology of rivers [3,4]. Installing multiple hydraulic weirs has a negative impact on aquatic ecosystems [5]. In addition, RRHSs reduce periodic hydrological disturbances in streams [6]. In the case of East Asia, where droughts and floods occur irregularly, RRHSs are a factor that makes river management more difficult [7].

Fragmentation impacts on rivers from weir construction and damming have been determined. Weir and dam construction causes species isolation [8]. Habitat transformation caused by a reduction in river connectivity favors generalist species over more habitat-specific species [9]. Lateral structures affect zooplankton community changes in the dry season rather than the wet season [10]. Recently, studies on biological effects from the fragmentation of rivers have focused on both the pre- and post-construction period. In fishes, decreasing populations and a higher proportion of exotic species were reported [11]. For fish with strong mobility, the fragmentation of rivers can be overcome via ecological fish pathways [12]. The proportion of habitats that change with the seasons decreased [13]. The dispersal of aquatic organisms along the continuum of rivers has also been studied [14,15].

Recently, zooplankton metacommunities along with spatiotemporal factors of rivers have been studied [10]. In the case of run-of-river damming in tropical regions, the flood pulse in pre-/post-dam periods was the main effector for the zooplankton community [16]. However, previous studies on the connectivity of rivers using zooplankton have several disadvantages, such as a short investigation period [1,17,18]. Due to the characteristics of zooplankton showing an unpatched distribution [19], data interpretation also could be biased. In the original connected river, the process of discovering the community similarity between upstream and downstream is required first. Therefore, to overcome the above limitations in this study, we used long-term monitoring data, especially only zooplankton population data, since zooplankton are a representative that reflect environmental factors [7,20]. In our study, we focused on the characteristics of zooplankton communities.

Zooplankton populations can be maintained in rivers despite them being unable to counteract unidirectional water flow. This is what is known as the drift paradox [21]. Therefore, zooplankton communities form biological gradients along the flow direction [22]. However, RRHSs have properties similar to lakes in terms of the periodicity of plankton dynamics, like clear-water phases [1]. Due to weir construction, the residence time of the waterbody is prolonged, which contributes to the outbreak of cyanobacteria communities [22,23].

Recently, to explain the response of organisms to environmental changes, approaches based on functional groups have been used, rather than basing them on existing taxa [24,25]. The classical biotic index based on population density and species composition, such as Shannon diversity, is not suitable for real-population responses in the pre-/post-dam period [11]. Functionally similar groups respond similarly to environmental changes [26,27]. Therefore, functional grouping using functional traits could utilize specific traits suitable for environmental changes.

South Korea is in Northeast Asia; it has a distinct summer monsoon and had an average annual rainfall of 1300 mm from 1990 to 2019. However, 54% of the rainfall is concentrated from June to August (https://www.index.go.kr (accessed on 1 September 2021); Statistics Korea Government Official Work Conference). Accordingly, over 18,000 reservoirs have been built nationwide to facilitate water use [28], and the weirs in large rivers are particularly important for water use in the dry season. Weirs are lateral structures installed on rivers that change the running water ecosystem into a lentic ecosystem (becoming an RRHS) and hinder water flow, which is a core process of the river ecosystem, thus changing the structure of the food web [11]. Catastrophically, South Korea's four major rivers have become RRHSs, which are made up of highly dense weirs with four large river projects (4LRPs) built from 2009 to 2011 as a national project [11,13]. Environmental and community changes induced by the 4LRPs have been investigated in many studies [11,13,22]. However, there are few studies on the deterioration of the connection between zooplankton communities induced by the 4LRPs. It widened the width of the rivers, increased the water depth, and slowed the flow rate. The transformed riparian zone, dredged sediment, and increased water volume can change habitat characteristics such as a dramatically decreasing population density in freshwater zooplankton [20,29]. Various factors need to be considered to understand the changes in zooplankton communities due to RRHSs [7,30], but understanding these factors can be challenging. Therefore, we considered only two factors: first, the body size of zooplankton, which simultaneously reflects the bottom-up effect leading to water quality–phytoplankton–zooplankton, and the top-down effect leading to fish–zooplankton. The predation pressure of the upper predator limits the length of zooplankton individuals and induces them to take refuge [7,31,32]. The main cause of shifts in zooplankton size distribution is the presence or absence of planktivorous fish [33–35]. To overcome predation pressure, zooplankton use refuges [36]. Representative refuges are substrates of the riparian zone. Therefore, the second factor was the swimming type (planktonic and epiphytic) that represents these habitat characteristics [20,36]. In addition, the swimming type is suitable for analyzing disturbances accompanying changes in flow rates and habitats [20].

This study investigated the changes in zooplankton communities through surveys conducted upstream and downstream of weirs on the Nakdong River, the longest river with the highest density of RRHSs in South Korea. The sites selected had all been monitored since before weir construction, rendering them useful for understanding the changes in zooplankton communities caused by RRHS. Specifically, we aimed to (1) confirm how the zooplankton community structure and composition changed before and after RRHS and (2) identify whether river connectivity is maintained through the time-series similarity of longitudinal zooplankton communities despite RRHS. Through this, we would like to confirm the changes in the zooplankton community caused by the anthropogenic freshwater ecosystem represented by RRHS.

## 2. Materials and Methods

### 2.1. Study Sites

The Nakdong River is the longest river in South Korea, located in the southeastern part of the Korean Peninsula in Northeast Asia (length: 525 km; river basin: 23,716.7 km$^2$). The Nakdong River has a gentle slope, and the elevation difference of the mainstream is less than 100 m. In the upper and middle streams, it flows along the tectonic line between the mesozoites, and in the downstream it flows through the porphyrite zone. The population of the mainstream and tributary basin is approximately 10 million (Figure 1). The annual rainfall is 1200 mm, and it has typical monsoon characteristics, with more than 60% of rainfall concentrated between June and September [37]. There are two dams, eight weirs, and one estuary bank in the mainstream of the Nakdong River, which is equivalent to a large reservoir every 58 km on average. The structure of the rivers also changed around the 4LRPs. At the most upstream survey point (Waegwan), the average water depth increased from 1.2 to 5.4 m and the river width expanded from 247 to 472 m. At the midpoint of the survey sites (Jukpo), the average water depth increased from 2.5 to 8.1 m and the river width expanded from 202 to 231 m. At the most downstream point (Mulgeum), the average water depth increased from 5.4 to 8.7 m and the river width expanded from 251 to 445 m [11].

There were six sites in this study (Figure 1). Five sites were points that could be compared up and down the weir (Waegwan-Goryeong, Goryeong-Jukpo, and Namji-Hanam), and four sites were points that could be compared between two consecutive points that exist between RRHS weirs (Jukpo-Namji and Hanam-Mulgeum). The survey was conducted on a bi-weekly basis (*n* = 761, 356, 372, 416, 386, and 417, respectively) from 2002 to 2008 (82 months) and 2012 to 2018 (80 months), and the survey data were converted into monthly averages.

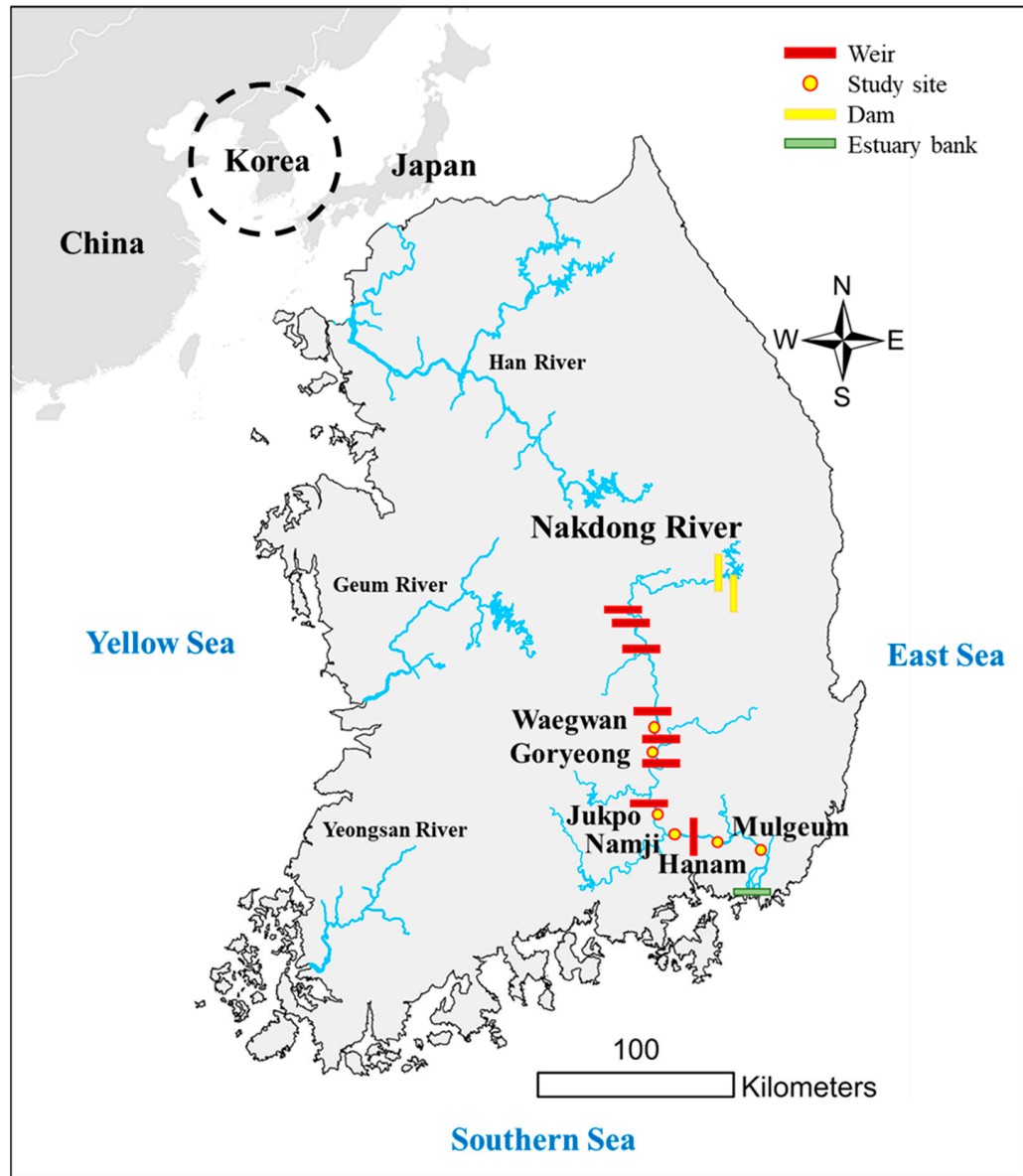

**Figure 1.** Description of study sites in the Nakdong River. The yellow dots are the study sites, and the red rectangles are the constructed weirs from the 4 large river projects (4LRPs) (coordinate system: PCS ITRF2000 TM).

### 2.2. Field Survey

Zooplankton samples were collected in 4 or 8 L water samples at 0.5 m depth of the riparian zone. The samples were filtered through a 32 μm nylon mesh and preserved in sugar formalin (4% for formaldehyde) [38]. As a result, the filtered water was concentrated to 100 mL. We extracted 1 mL using a homogeneous sample through inverting. The zooplankton samples were counted using an optical microscope (Zeiss Axiolab re; Carl Zeiss, Inc., New York, NY, USA) at × 40–100 magnification in a Sedgwick-Rafter chamber. Finally, counting data were converted to individuals per liter unit. Zooplankton taxa were identified at the genus or species level, except for nauplii and copepodites [39,40]. The zooplankton were categorized by taxon (rotifers, cladocerans, and copepods).

### 2.3. Data Analysis

A *t*-test in SPSS (version 26.0 for Windows; SPSS Inc., Chicago, IL, USA) was used to compare the mean of the zooplankton community (the number of species and popula-

tion density) before and after the 4LRPs. To examine zooplankton species for the actual ecological status, we performed k-means clustering using both length (µm) and swimming-type (planktonic, epiphytic, and mixing type) data via the elbow method for optimal clustering numbers using the "NbClust" package in R 4.1.0. (http://cran.r-project.org (accessed on 15 September 2021)) [41]. To do so, nominal data were assigned a number (planktonic: −1, epiphytic: 1, and mixing type: 0), and both factors were standardized and analyzed. To minimize seasonality, we performed the Mann–Kendall (MK) test using the seasonal decomposition in SPSS. The MK test was applied to assess the significance of zooplankton community composition trends as an average of monthly values of the survey sites using the "trend" package in R [42]. As zooplankton communities are not uniformly distributed [19], a simple and robust MK test was appropriate for analyzing our non-parametric data [43]. Dynamic time warping (DTW) was used to confirm similar pattern changes in the zooplankton community using monthly data from each site in the six sites of the Nakdong River. DTW is a time series alignment technique that was first applied for spoken word recognition [44]. Hierarchical clustering analysis was used to examine the relationship among survey sites along with the RRHS period using Ward's method and Euclidean distance. These analyses were performed using the "dtwclust" package [45]. All analyses were performed using R 4.1.0. software (http://cran.r-project.org (accessed on 18 September 2021)).

## 3. Results

### 3.1. Time-Series Fluctuations of Zooplankton

In total, 164 species (SP) were identified, comprising 119 rotifers, 33 cladocerans, and 12 copepods. The average population density (PD) at the six sites was 751.1 ind./L. The SP number and PD fluctuated every year (Figure 2). The natural river had a lower SP number and PD values than the RRHS river (Figure 2). The average SP number collected before (9.5 ± 0.6) and after (10.2 ± 0.4) the RRHS construction was similar; however, the average PD increased by more than four times. Specifically, PD increased by 3.9 times for rotifers, 8.7 times for cladocerans, and 10.2 times for copepods. RRHS impacted zooplankton community composition. In the lotic Nakdong River, 137 SP were identified. In the RRHS Nakdong River, 104 SP were identified. In total, there were 22 new species of zooplankton; overall, the SP number decreased by 33. The most common species were *Polyarthra vulgaris*, *Keratella cochlearis*, and *Synchaeta* sp.

As a result of the elbow method to determine the optimal number of clusters, the zooplankton species were classified into three groups according to their length and swimming type (Table 1). Cluster 1 consisted of large planktonic species. Cluster 2 consisted of small and epiphytic species. Cluster 3 consisted of small and planktonic species.

**Table 1.** Characteristics of the zooplankton community classified according to length and swimming type using the elbow method and k-means clustering.

| Category | | Cluster 1 | Cluster 2 | Cluster 3 |
|---|---|---|---|---|
| Species | Rotifers | 3 | 63 | 53 |
| | Cladocerans | 12 | 10 | 11 |
| | Copepods | 8 | 2 | 3 |
| | Total | 23 | 75 | 67 |
| Swimming type | Planktonic | 19 | 0 | 67 |
| | Mixing | 3 | 23 | 0 |
| | Epiphytic | 1 | 52 | 0 |
| | Total | 23 | 75 | 67 |
| Length (µm) | Minimum | 733 | 50 | 70 |
| | Maximum | 2000 | 810 | 700 |

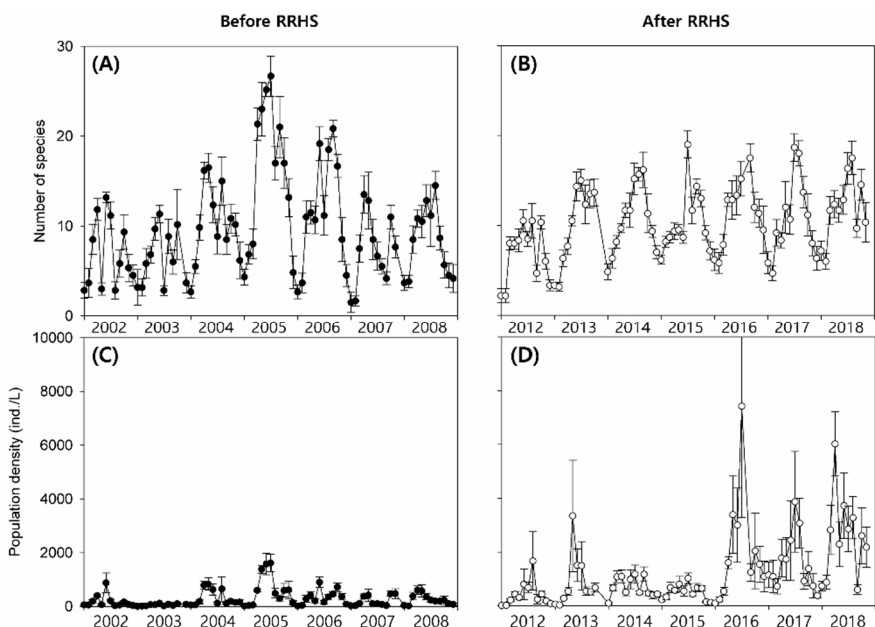

**Figure 2.** Monthly variation of the zooplankton community before and after the construction of the river–reservoir hybrid system. (SP of zooplankton before (**A**) and after (**B**) RRHS; PD of zooplankton before (**C**) and after (**D**) RRHS).

Although rotifers had the largest proportion of species regardless of RRHS, the SP number and PD ratios of cladocerans and copepods increased after the RRHS (Figure 3). The average SP did not show a significant difference ($p = 0.409$), whereas total PD did show a significant difference ($p < 0.001$). Rotifer SP did not show a significant difference ($p = 0.470$). The PD of the three taxonomic groups and the SP number of both cladocerans and copepods showed significant differences between pre- and post-4LRPs ($p < 0.001$). After construction, the population density of cladocerans increased relative to that of other taxa (2012–2014), but at the end of the survey, rotifers again dominated. Clusters 2 and 3 occupied most of both SP and PD in a similar proportion (Figure 4). In detail, the SP of cluster 1 showed a significant difference ($p < 0.001$). The PD of the three clusters showed significant differences between pre- and post-4LRPs ($p < 0.001$). Overall, rotifer-dominant clusters (clusters 1 and 2) did not show significant differences in SP.

Each group showed an increasing trend during both periods (Table 2). The zooplankton community showed a tendency in the Nakdong River before the disturbance. The SP of large cladocerans showed a significant increase (cladoceran: $p < 0.001$, Z = 3.404; cluster 1: $p = 0.017$, Z = 2.393). There was also an increase in PD (cladoceran: $p < 0.001$, Z = 4.007; cluster 1: $p = 0.034$, Z = 2.124). The Nakdong River after the RRHS showed more clearly the increase/decrease direction of the zooplankton community compared to the natural state. The overall SP number and PD increased (SP: $p < 0.001$, Z = 3.586; PD: $p < 0.001$, Z = 5.738). The main groups that affected this were rotifers (SP: $p < 0.001$, Z = 4.936; PD: $p < 0.001$, Z = 7.075) and cluster 2 (SP: $p < 0.001$, Z = 5.248; PD: $p < 0.001$, Z = 6.278).

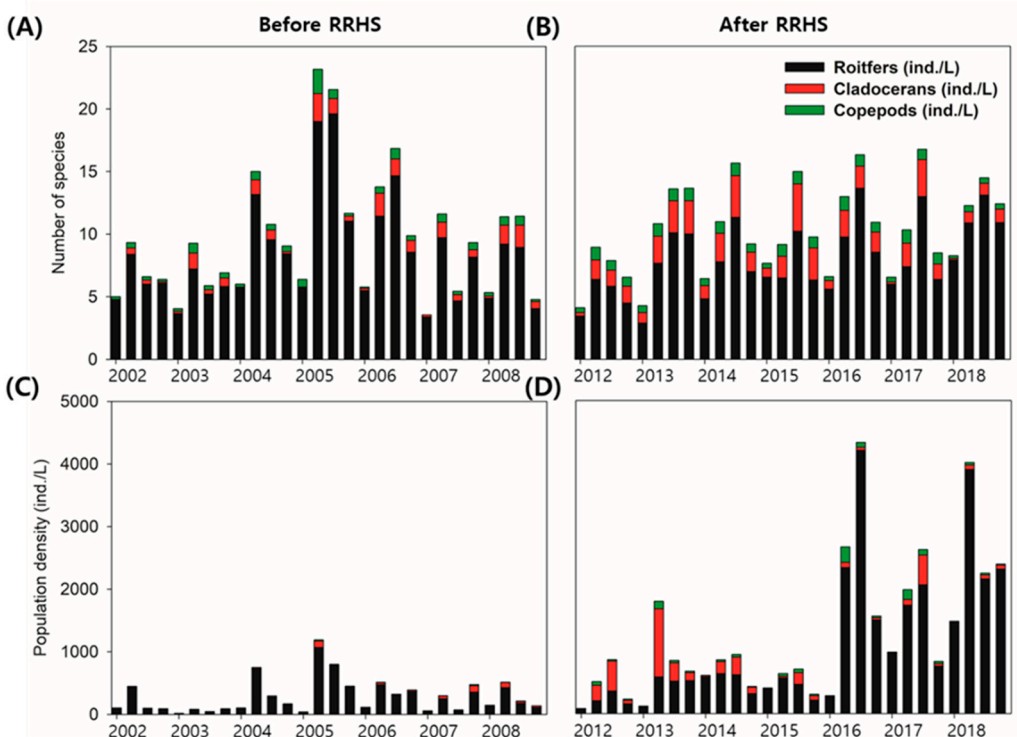

**Figure 3.** Taxonomic zooplankton community before and after the construction of the river–reservoir hybrid system. (SP of zooplankton before (**A**) and after (**B**) RRHS; PD of zooplankton before (**C**) and after (**D**) RRHS).

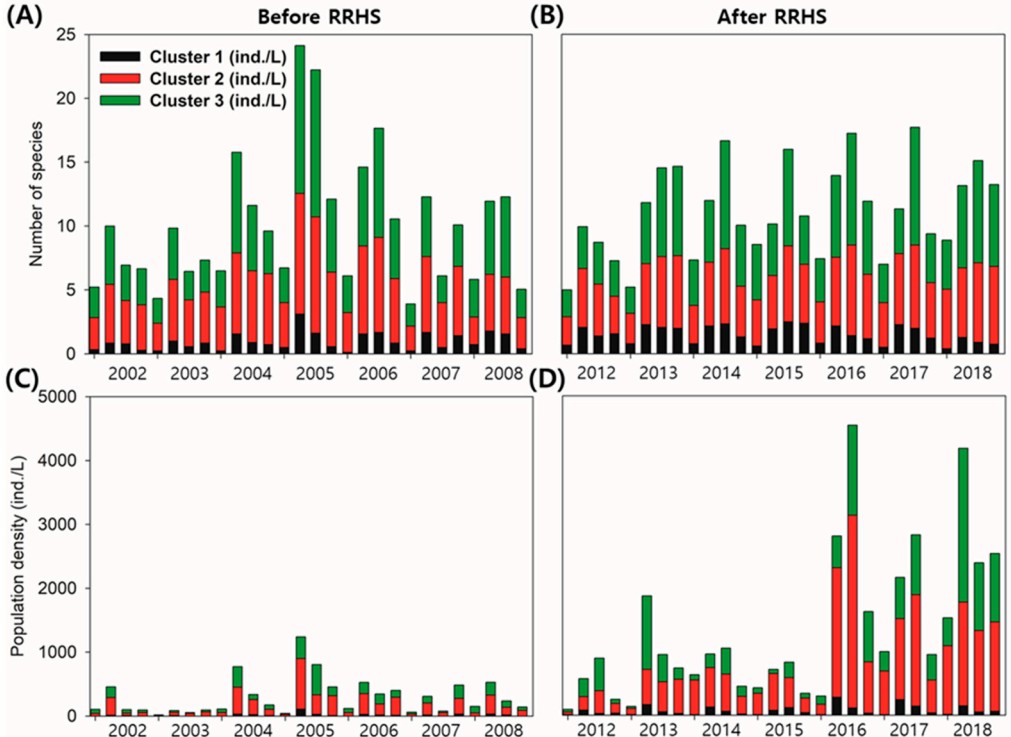

**Figure 4.** Functional zooplankton community before and after the construction of the river–reservoir hybrid system. (Cluster 1 consisted of large planktonic species. Cluster 2 consisted of small and epiphytic species. Cluster 3 consisted of small and planktonic species). (SP of zooplankton before (**A**) and after (**B**) RRHS; PD of zooplankton before (**C**) and after (**D**) RRHS).

**Table 2.** Results of the Mann–Kendall trend test after seasonal decomposition of the zooplankton community dataset (positive Z-value: increasing trend, negative Z-value: decreasing trend).

| Category | | Before RRHS | | After RRHS | |
|---|---|---|---|---|---|
| | | $p$ | Z Value | $p$ | Z Value |
| Species | Total | 0.325 | 0.985 | 0.000 | 3.586 |
| | Rotifers | 0.597 | 0.529 | 0.000 | 4.936 |
| | Cladocerans | 0.001 | 3.404 | 0.027 | −2.215 |
| | Copepods | 0.327 | 0.981 | 0.025 | −2.237 |
| | Cluster 1 | 0.017 | 2.393 | 0.060 | −1.878 |
| | Cluster 2 | 0.848 | 0.192 | 0.000 | 5.248 |
| | Cluster 3 | 0.262 | 1.121 | 0.006 | 2.726 |
| Population density | Total | 0.075 | 1.778 | 0.000 | 5.738 |
| | Rotifers | 0.197 | 1.289 | 0.000 | 7.075 |
| | Cladocerans | 0.000 | 4.007 | 0.004 | −2.900 |
| | Copepods | 0.271 | 1.102 | 0.589 | 0.540 |
| | Cluster 1 | 0.034 | 2.124 | 0.083 | 1.733 |
| | Cluster 2 | 0.056 | 1.914 | 0.000 | 6.278 |
| | Cluster 3 | 0.115 | 1.578 | 0.000 | 5.497 |

### 3.2. Time-Series Trends and Longitudinal Patterns

The similarity in the time-series changes of zooplankton communities appearing at the six survey points showed differences before and after RRHS (Figure 5). In the Nakdong River before RRHSs, the change in SP according to the flow of the water was similar (except for Mulgeum), but after controlling the flow of the water body, the upstream and downstream similarities disappeared. Similarly, PD was randomly arranged with time-series similarities between the longitudinal points from upstream to downstream.

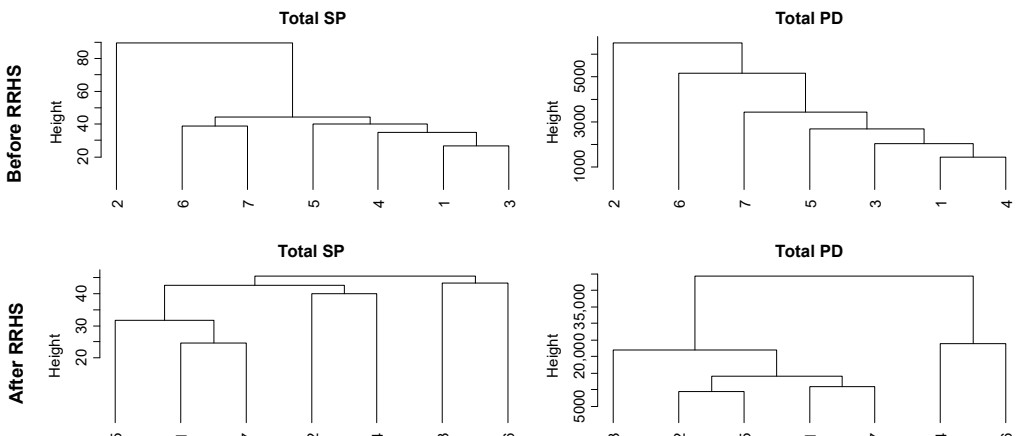

**Figure 5.** Dendrograms of number of species (SP) and population density (PD) in the Nakdong River using dynamic time warping. The analysis was performed using Ward's method and Euclidean distance as the measure of similarity (total: overall zooplankton community, 1: average of 6 sites, 2: Mulgeum, 3: Hanam, 4: Namji, 5: Jukpo, 6: Goryeong, and 7: Waegwan).

Prior to the RRHS, connectivity was evident in the SP number and PD of the rotifers as the dominant group. However, after the RRHS, all connectivity was disrupted in the taxonomic groups (Figure 6). The functional groups showed different patterns. Regardless of the RRHS, the SP number was the highest in cluster 3 and PD was the highest in cluster 2. These were the dominant factors in SP (cluster 3) and PD (cluster 2). In the connected river, the height of cluster 2 with the highest PD showed connectivity for both SP and PD, but the height of cluster 3 with the highest SP number showed a dendrogram order similar to the upstream and downstream order of SP only.

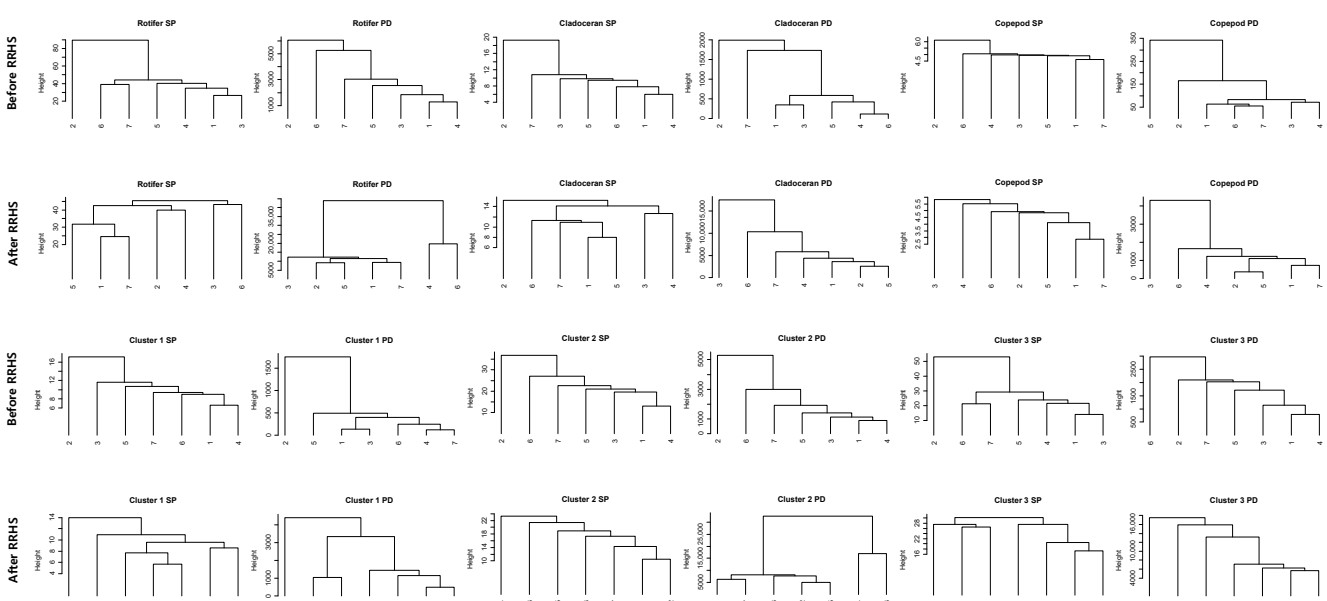

**Figure 6.** Detailed dendrograms of the number of species (SP) and population density (PD) in the Nakdong River using dynamic time warping. Analysis was performed using Ward's method and Euclidean distance as the measure of similarity (1: average of 6 sites, 2: Mulgeum, 3: Hanam, 4: Namji, 5: Jukpo, 6: Goryeong, and 7: Waegwan).

## 4. Discussion

### 4.1. Importance of Long-Term Zooplankton Data

Researchers have conducted many studies to identify and predict changes in ecosystems [20,46] and used experimental data from the field or laboratory to support their findings. Long-term data are useful for these situations. If there are insufficient data from before the disturbance, the environmental change cannot be determined. Ecosystem changes such as the 4LRPs are inevitable considering the anthropogenic disturbance of dam construction. Regardless of dam size, planktonic and riparian taxa of rotifers increase after decreased water velocity [47]. Our results confirmed that the zooplankton population increased regardless of the connectivity of the Nakdong River. Therefore, long-term data of previous uncontrolled ecosystems can be used to evaluate the current natural or anthropogenic ecosystem or predict future ecosystems [48]. In addition, long-term monitoring is important for the development of empirical academic and management policies, because ecosystem changes can be quantified as ecological responses [49–51].

Although long-term data are valuable for identifying ecosystem transitions, it is important to consider which parameters to monitor [49]. To overcome resource and time constraints, using indicator species has been proposed. However, there are many taxa that can be indicators [52]. Combing taxa into functional groups suitable for the purpose of the specific study has advantages in data interpretation [20]. In our results, habitat alteration and partial waterflow discontinuity by the 4LRPs changed the composition and structure of zooplankton according to the clusters.

Using zooplankton as indicator species has advantages in terms of a short life cycle and adaptation ability from disturbances [53,54]. Assembling zooplankton into functional groups effectively represents the biological components and disturbances of ecology [20,55]. In our results, the two selected zooplankton traits were adequate to represent the before-and-after comparison of the 4LRPs in the cluster 2 SP/PD and cluster 3 SP (Figure 6). Time-series similarity in the zooplankton community change was found between sites close in the connected Nakdong River. Through this, it can be inferred that the connectivity between upstream and downstream was damaged by the RRHS. Mulgeum especially, the lowest survey point, represented the unique zooplankton dynamics of the Nakdong River. This is because the area is continuously managed as a water intake source that supplies

water to Busan Metropolitan City, inhabited by more than 3 million people. Considering such activities in rivers, representative points of river monitoring should be established.

*4.2. River–Reservoir Hybrid System with Zooplankton*

River–reservoir hybrid systems reduce periodic hydrological disturbances in streams [6]. RRHSs have properties similar to those of lakes in terms of periodicity and plankton dynamics [1]. Due to the 4LRPs, the Nakdong River had stagnant and volumetrically increased waterbodies. As a result, the suitability of the Nakdong River for zooplankton habitats increased. The overall PD of zooplankton in the Nakdong River showed a stronger trend after the RRHS. In the case of SP, the average SP number showed an increasing trend despite decreased accumulated SP. This indicates that the connectivity between longitudinal points of the river had weakened and meant that the frequency of occurrence of certain species had gradually increased. In conclusion, partially opening a sluice gate or artificial sluice gate control cannot ensure connectivity between weir sections. A better environment for zooplankton provides only a proliferation opportunity for species that have adapted to it, leading to their dominance.

**5. Conclusions**

Long-term monitoring results performed at six longitudinal survey points of the river for 14 years helped to understand how the similarity of zooplankton communities changes due to the construction of the lateral structures in the Nakdong River. A stagnant and increased water body caused by RRHSs functioned as a zooplankton habitat, but the longitudinal gradient, a unique characteristic of zooplankton in running water, had disappeared. Significant changes have also occurred in the patterns of previously thriving zooplankton taxa communities. Large-scale anthropogenic disturbances in rivers, such as the 4LRPs, should be thoroughly evaluated before and after construction and managed with constant advice from experts in various fields. Nevertheless, the biodiversity of rivers will be impacted. Although minimizing disturbances should be the primary goal, zooplankton can be a useful indicator of river ecosystem health.

**Author Contributions:** Conceptualization, E.-J.K. and G.-J.J.; methodology, E.-J.K. and Y.D.; software, Y.D.; validation, Y.D. and H.J.; formal analysis, E.-J.K. and E.J.; investigation, E.-J.K., E.J. and H.J.; resources, H.-W.K.; data curation, H.-W.K.; writing—original draft preparation, E.-J.K. and E.J.; writing—review and editing, E.-J.K. and H.J.; visualization, E.-J.K.; supervision, H.J.; project administration, G.-J.J. and H.-W.K.; funding acquisition, G.-J.J. All authors have read and agreed to the published version of the manuscript.

**Funding:** This study was supported by the BK21 Four Program of the Pusan National University. This research was funded by the National Research Foundation of Korea (grant number NRF-2020R1C1C1009066).

**Institutional Review Board Statement:** Not applicable.

**Informed Consent Statement:** Not applicable.

**Data Availability Statement:** The data presented in this study are available on request from the corresponding author.

**Acknowledgments:** This research was funded by the National Research Foundation of Korea (grant number NRF- 2020R1C1C1009066). We thank the Freshwater Ecology Lab members at Pusan National University who shared their ecological knowledge of limnology, especially in the Nakdong River basins.

**Conflicts of Interest:** The authors declare no conflict of interest.

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
