# Peer review of "Impact of River-Reservoir Hybrid System on Zooplankton Community and River Connectivity"

_sustainability, doi:10.3390/su14095184_

Round 1

Reviewer 1 Report

This paper analyses the effect of hydrological alteration caused by weirs in the zooplankton community of a Korean river. It contributes to expand the knowledge of the impact of different man-made structures in the riverine ecosystems, which is not well understood. The research question and methodological design are clearly stated and the results are adequately discussed, but there are some issues that need further development, explanation and clarification.

The most pressing issue is the writing: The text lacks coherence, jumping between badly connected sentences (e.g. lines 57-63) with no visible organization inside paragraphs. Moreover, the introduction needs more development to appropriately include the research in its context, adding more references in certain places (e.g. line 36 about the effect of barriers in fish populations). I am sure there are more studies about the effects of dams in zooplankton that can be referenced. Their results need to be better explained as well as the place of this research in that context. Where does this research fit? What are its main contributions?

In general, the text is very difficult to read with numerous grammatical and syntax errors, mixing of past and present tenses and poor sentence cohesion. It needs to be revised in depth and rewritten by someone fluid in the English language to improve its quality.

The abstract is confusing. It also needs rewriting and expansion.  

The discussion could be more elaborated, with more connections to other works and more references. The results may be further debated citing other similar research as well as the potential drawbacks of this study.

Some methods also need further clarification. I detail more specific comments below:

Line 25: more keywords.

Line 30: Describe RRHSs in depth and why they are unique and important to study. The name is not widely used as I have not found further mentions to them. The references 2, 3, 4 and 5 do not clearly support the existence and characteristics attributed to them.

Line 31: RRHSs are not built to support aquatic ecosystems, only to serve human needs, please reword or remove that.

Line 35: Fragmentation instead of discontinuity.

Line 38: Add more information and more references on the different effects of barriers on river ecosystems. There are many studies of the effect on fish, sediment transport, geomorphology, dispersion, etc. The whole paragraph is very difficult to understand with weird sentence structure and cohesion.

Line 47: Add references on the disadvantages of previous studies.

Line 51: Add references about studies “only after dam installation”.

Line 54: Add a reference to that statement: “zooplankton is a representative that reflects all environmental factors”.

Line 85: riparian may be a better word than littoral, that refers to the coast.

Line 87: Add more references that 26.

Line 98: Add a reference that supports that statement that swimming type is suitable for that analysis.

Line 117: What are the 4LRPs? I only see three studied weirs.

Line 123: Are the impact-control analyses? Or only before-after dam construction? Is the habitat similar in all study sites? Describe the differences between sites. The analysis appear to be done using the means of all the sites, is that correct?

Line 131: Figure 1. Point to the 4LRPs.

Line 134: More detail in the collection and analysis of samples. How much of the 4/8 l samples were filtered and observed under the microscope?

Line 142: what does “mean of zooplankton community” means? Population size? Biomass? Are parametric assumptions met?

Line 148: Mann-Kendall test still needs independent samples. What is the autocorrelation of the time series and between sites? If the independence assumption is not met, then other analysis should be performed to account for temporal autocorrelation.

Line 162: the species numbers sum 164, not 165.

Line 165: more appropriate to say free or natural river, not “continuing”.

Line 176: Table 1. The method is the clustering. The elbow method is only a tool to select the most adequate number of clusters.

Line 212: The table caption is too short. Expand it more.

Author Response

  First of all, the co-authors and I would like to appreciate to reviewers and editor for their kind response and comments to improve our manuscript draft (sustainability-1612523). According to the comments, we would like to submit a written reply. Responses (corrections) for Reviewer 1 are tracking by word program.

Reviewer 1

â–ª This paper analyses the effect of hydrological alteration caused by weirs in the zooplankton community of a Korean river. It contributes to expand the knowledge of the impact of different man-made structures in the riverine ecosystems, which is not well understood. The research question and methodological design are clearly stated and the results are adequately discussed, but there are some issues that need further development, explanation and clarification.

The most pressing issue is the writing: The text lacks coherence, jumping between badly connected sentences (e.g. lines 57-63) with no visible organization inside paragraphs. Moreover, the introduction needs more development to appropriately include the research in its context, adding more references in certain places (e.g. line 36 about the effect of barriers in fish populations). I am sure there are more studies about the effects of dams in zooplankton that can be referenced. Their results need to be better explained as well as the place of this research in that context. Where does this research fit? What are its main contributions?

In general, the text is very difficult to read with numerous grammatical and syntax errors, mixing of past and present tenses and poor sentence cohesion. It needs to be revised in depth and rewritten by someone fluid in the English language to improve its quality.

The abstract is confusing. It also needs rewriting and expansion.  

The discussion could be more elaborated, with more connections to other works and more references. The results may be further debated citing other similar research as well as the potential drawbacks of this study.

==> Thank you for the comments. We would revise a manuscript following your comments. Maurice JM Lineman is a native Canadian and have revised this article before submission.

â–ª Line 25: more keywords.

==>  We added adequate three keywords in the abstract part.

Keywords: dynamic time warping; longitudinal connectivity, weir effect, long-term monitoring, Nakdong River

â–ª Line 30: Describe RRHSs in depth and why they are unique and important to study. The name is not widely used as I have not found further mentions to them. The references 2, 3, 4 and 5 do not clearly support the existence and characteristics attributed to them.

==> Sentence was removed. And we added an adequate sentence.

As RRHSs increase the stability of water bodies, from a human perspective, there is substantial incentive to construct them. In the case of East Asia, where droughts and floods occur irregularly, the RRHS is a factor that makes river management more difficult.

â–ª Line 31: RRHSs are not built to support aquatic ecosystems, only to serve human needs, please reword or remove that.

==> We added additional sentences after that sentence.

However, lateral structures have been found to affects limnology and hydrology of rivers. Installing multiple hydraulic weirs has a negative impact on aquatic ecosystems.

â–ª Line 35: Fragmentation instead of discontinuity.

==> We replaced the word.

â–ª Line 38: Add more information and more references on the different effects of barriers on river ecosystems. There are many studies of the effect on fish, sediment transport, geomorphology, dispersion, etc. The whole paragraph is very difficult to understand with weird sentence structure and cohesion.

==> We wrote some sentences of the paragraph to make them easier to understand. And we split one paragraph into two.

Fragmentation impacts on rivers by weir construction and damming have been determined. Lateral structures affected zooplankton community changes in the dry season rather than the wet season. Recently, biological effects from fragmentation of river have focused on both the pre- and post-construction period. Decreasing overall population and leading to higher proportions of exotic species were happened in fishes. For fish with strong mobility, Fragmentation of the river can be overcome via eco-logical fish pathways. The proportion of habitats that changes with the seasons had decreased. Dispersal of aquatic organisms along the continuum of rivers has also been studied.

Recently, zooplankton metacommunities along with spatiotemporal factors of rivers have been studied. In the case of run-of-river damming in tropical regions, the flood pulse in pre/post-dam periods was the main effector to the zooplankton community. However, previous studies of the connectivity of rivers using zooplankton have several disadvantages, such as a short investigation period. Due to the characteristics of zooplankton showing an unpatched distribution, data interpretation also could be biased. in the original connected river, the process of dis-covering the community similarity between upstream and downstream is required first. Therefore, to overcome above limitation in this study, we used long-term monitoring data, especially only zooplankton population data since zooplankton is a representative that reflects environmental factors. In our study, we focused on the characteristics of zooplankton communities.

â–ª Line 47: Add references on the disadvantages of previous studies.

==> We added two references.

Kim, H.W.; Joo, G.J. The longitudinal distribution and community dynamics of zooplankton in a regulated large river: a case study of the Nakdong River (Korea). Hydrobiologia, 2000, 438, 171-184.

Kamboj, V.; Kamboj, N. Spatial and temporal variation of zooplankton assemblage in the mining-impacted stretch of Ganga River, Uttarakhand, India. Environ. Sci. Pollut. Res. 2020, 27, 27135-27146.

â–ª Line 51: Add references about studies “only after dam installation”.

==> To rewrite the paragraph, the sentence was removed.

Most previous studies analyzed a spatial difference in zooplankton community structure between longitudinal sites using only after a dam installation data.

â–ª Line 54: Add a reference to that statement: “zooplankton is a representative that reflects all environmental factors”.

==> We revised the sentence and added references.

Therefore, to overcome above limitation in this study, we used long-term monitoring data, especially only zooplankton population data since zooplankton is a representative that reflects environmental factors.

Ko, E.J.; Kim, D.K.; Jung, E.S.; Heo, Y.J.; Joo, G.J.; Kim, H.W. Comparison of Zooplankton Community Patterns in Relation to Sediment Disturbances by Dredging in the Guemho River, Korea. Water 2020, 12, 3434.

Kim, D.K.; Jeong, K.S.; Chang, K.H.; La, G.H.; Joo, G.J.; Kim, H.W. Patterning zooplankton communities in accordance with annual climatic conditions in a regulated river system. Nakdong River, South Kores). Int. Rev. Hydrobiol. 97, 55–72 (2012).

â–ª Line 85: riparian may be a better word than littoral, that refers to the coast.

==> We replaced the corresponding word throughout the manuscript.

â–ª Line 87: Add more references that 26.

==> We added one reference.

Zhang, S.; Zhou, Q.; Xu, D.; Lin, J.; Cheng, S.; Wu, Z. Effects of sediment dredging on water quality and zooplankton com-munity structure in a shallow of eutrophic lake. J. Environ. Sci. 2010, 22, 218-224.

â–ª Line 98: Add a reference that supports that statement that swimming type is suitable for that analysis.

==> We added one reference.

Ko, E.J.; Kim, D.K.; Jung, E.S.; Heo, Y.J.; Joo, G.J.; Kim, H.W. Comparison of Zooplankton Community Patterns in Relation to Sediment Disturbances by Dredging in the Guemho River, Korea. Water 2020, 12, 3434.

â–ª Line 117: What are the 4LRPs? I only see three studied weirs.

==> Although the 4LRPs were projects that were carried out on four major rivers, we had shown only the Nakdong River as our study site.

â–ª Line 123: Are the impact-control analyses? Or only before-after dam construction? Is the habitat similar in all study sites? Describe the differences between sites. The analysis appear to be done using the means of all the sites, is that correct?

==> We performed the analysis using zooplankton analyzed with samples collected in the field without any control factors.

==> Six study sites were similar habitats with about 1 m depth where aquatic plants and stones coexist.

==> Yes, it was correct. Survey data were converted into monthly averages. When showing the trend of long-term data, the average of survey sites was used, but in the case of DTW analysis, each site was separated.

â–ª Line 131: Figure 1. Point to the 4LRPs.

==> We modified Figure 1 and added a description of it.

Figure 1. Description of study sites in the Nakdong River. The yellow dots are the study sites, and the red rectangles are the constructed weirs from 4 large river projects (4LRPs).

â–ª Line 134: More detail in the collection and analysis of samples. How much of the 4/8 l samples were filtered and observed under the microscope?

==> Thank you for your comments. We wrote more detail description about zooplankton sampling and counting process.

Zooplankton samples were collected in 4 or 8 L water samples at 0.5 m depth of riparian zone. The samples were filtered through a 32 µm nylon mesh and preserved in sugar formalin (4% for formaldehyde). As a results, the filtered water was con-centrated to 100 ml. We extracted 1 ml using a homogeneous sample through inverting. The zooplankton samples were counted using an optical microscope (Zeiss Axiolab re; Carl Zeiss, Inc.) at x 40–100 magnification in a Sedgwick-Rafter chamber. Finally, counting data were converted to individuals per liter unit. Zooplankton taxa were identified at the genus or species level, except for nauplii and copepodites. The zooplankton were categorised by taxon (rotifers, cladocerans, and copepods).

â–ª Line 142: what does “mean of zooplankton community” means? Population size? Biomass? Are parametric assumptions met?

==> An explanation has been added to help the understanding of the sentence.

A t-test in SPSS (version 26.0 for Windows; SPSS Inc.) was used to compare the mean of zooplankton community (the number of species and population density) be-fore and after 4LRPs.

==> Since the number of samples exceeds 100, a t-test was performed.

â–ª Line 148: Mann-Kendall test still needs independent samples. What is the autocorrelation of the time series and between sites? If the independence assumption is not met, then other analysis should be performed to account for temporal autocorrelation.

==> Thank you for your comments. To exclude autocorrelation, the MK test was re-performed using a data set after seasonal decomposition. To minimize seasonality, we performed the Mann-Kendall (MK) test using the seasonal decomposition data in SPSS.

â–ª Line 162: the species numbers sum 164, not 165.

==> We replaced the word.

â–ª Line 165: more appropriate to say free or natural river, not “continuing”.

==> We replaced the word.

â–ª Line 176: Table 1. The method is the clustering. The elbow method is only a tool to select the most adequate number of clusters.

==> We tried to classify zooplankton groups through K-means clustering analysis, and the elbow method was used to find the optimal number of groups.

â–ª Line 212: The table caption is too short. Expand it more.

==> We revised the sentence.

Results of Mann-Kendall trend test after seasonal decomposition data set of zooplankton commu-nity. (positive Z value: increasing trend, negative Z value: decreasing trend)

Reviewer 2 Report

I have read with interest your paper on Impact of river-reservoir hybrid system on zooplankton community and river connectivity. The structure of the article is right and the ideas clearly presented in general.

Results presented in chapter 3.1 refer to time variability of characteristics from all study sites in one (all the data (average) in one graph – for example: fig 2, 3, 4). Analysis of the data structure and tendencies in periods before and after RRHS construction is very interesting. However, in the period after RRHS, the weir is not the only one factor which might affect the zooplankton community in a new way. There might be changes in other factors determining zooplankton structure between these two periods like: biological, ecological, climatological or changes in anthropogenic impact (agricultural structure, water management etc.). As a result, it can not be clearly proved which part of the observed changes in zooplankton is determined by dam construction.

The solution of this problem is quite simply, especially the authors have at disposal a big and detailed data set. Let’s take for example two study points. Namji located upstream and Hanam located downstream of the weir (Gwangsimjeong?). If theoretically (I do not know the real data), in upstream site average number of SP was 10 before RRHS and after that it was equal 12, the increase of species number is equal 2. If in downstream site this increase between periods is equal 5, we may say that the weir determined the increase of SP number equal 3, because of the difference between downstream and upstream site. Such assumption might be applied to the other characteristics and variability parameters. An example of such kind of analysis in relation to low flows might be found in: https://doi.org/10.2478/limre-2021-0006

It is a pity that authors did not exploit the huge potential of their data set. The article is a bit too short and too synthetic. I suggest considering such approach in a part of manuscript

The conclusions section needs to be expanded and improved. It should contain more links with the paper. It also should explain which of investigated processes and changes are exactly determined by RRHS

In line 216, the context of ‘natural flow’ should be explained

In the last position of references [53] authors’ names have to be corrected: ‘KrzysztoÅ„, W.; Kosiba, J.’ instead of ‘Wojciech, K.; Joanna, K.’

Author Response

  First of all, the co-authors and I would like to appreciate to reviewers and editor for their kind response and comments to improve our manuscript draft (sustainability-1612523). According to the comments, we would like to submit a written reply. Responses (corrections) for Reviewer 2 are tracking by word program.

â–ª I have read with interest your paper on Impact of river-reservoir hybrid system on zooplankton community and river connectivity. The structure of the article is right and the ideas clearly presented in general.

Results presented in chapter 3.1 refer to time variability of characteristics from all study sites in one (all the data (average) in one graph – for example: fig 2, 3, 4). Analysis of the data structure and tendencies in periods before and after RRHS construction is very interesting. However, in the period after RRHS, the weir is not the only one factor which might affect the zooplankton community in a new way. There might be changes in other factors determining zooplankton structure between these two periods like: biological, ecological, climatological or changes in anthropogenic impact (agricultural structure, water management etc.). As a result, it can not be clearly proved which part of the observed changes in zooplankton is determined by dam construction.

The solution of this problem is quite simply, especially the authors have at disposal a big and detailed data set. Let’s take for example two study points. Namji located upstream and Hanam located downstream of the weir (Gwangsimjeong?). If theoretically (I do not know the real data), in upstream site average number of SP was 10 before RRHS and after that it was equal 12, the increase of species number is equal 2. If in downstream site this increase between periods is equal 5, we may say that the weir determined the increase of SP number equal 3, because of the difference between downstream and upstream site. Such assumption might be applied to the other characteristics and variability parameters. An example of such kind of analysis in relation to low flows might be found in: https://doi.org/10.2478/limre-2021-0006.

It is a pity that authors did not exploit the huge potential of their data set. The article is a bit too short and too synthetic. I suggest considering such approach in a part of manuscript.

The conclusions section needs to be expanded and improved. It should contain more links with the paper. It also should explain which of investigated processes and changes are exactly determined by RRHS.

==> Thank you for the comments. I read the article you recommended with interest. Unlike the situation in Poland, the total water volume of RRHSs in South Korea highly increased regardless of the upstream and downstream of the Nakdong River through damming. This was due to the construction of an estuary bank of 525 km of water in the Nakdong River, installing high-density dams, and river expansion constructions.

==> We also revised conclusion part.

Long-term monitoring results performed at 6 longitudinal survey points of the river for 14 years helped to understand how the similarity of zooplankton community changes due to the construction of the lateral structures in Nakdong River. A stagnant and increased water body caused by RRHSs functions as a zooplankton habitat, but the longitudinal gradient, a unique characteristic of zooplankton in running water, had disappeared. Significant changes have also occurred in the patterns of previously thriving zooplankton taxa communities. Large-scale anthropogenic disturbances in rivers, such as 4LRPs, should be thoroughly evaluated before and after construction and managed with constant advice from experts in various fields. Nevertheless, the biodiversity of rivers will be impacted. Although minimizing disturbances should be the primary goal, zooplankton can be a useful indicator of river ecosystem health.

In line 216, the context of ‘natural flow’ should be explained.

==> We revised the sentence.

In the Nakdong River before RRHSs, the change in SP according to the flow of water was similar (except for Mulgeum), but after controlling the flow of the water body, the upstream and downstream similarities disappeared.

In the last position of references [53] authors’ names have to be corrected: ‘KrzysztoÅ„, W.; Kosiba, J.’ instead of ‘Wojciech, K.; Joanna, K.’

==> We revised the sentence.

Krzysztoń, W.; Kosiba, J. Variations in zooplankton functional groups density in freshwater ecosystems exposed to cyano-bacterial blooms. Sci. Total Environ. 2020, 730, 139044.

Reviewer 3 Report

The manuscript describes the possible longitudinal changes of zooplankton communities in a river-reservoir hybrid system, that is a particular transition environment, due to anthropogenic modification for the construction of some weirs along the Nakdong River. The monitored zooplankton community trends in the 2002-2018 period could allow a better understanding of the influences of these manmade structures on river connectivity, as declared in the two main objectives of this research.

Even if I am not an ecologist but a transition environments geomorphologist with expertise on dam effects on the landscape, however, I noted some points and a figure to improve so that help the not Asian reader to better understand the study area features.

Generally, the manuscript is quite well structured: the Introduction, Materials and Methods, Results, and Discussion sections are congruent and sufficient. The Conclusions section is very poor, only six lines (...): I suggest extending it summarizing better the main results.

References are sufficient (53) and pertinent, but a few more citations are suggested (see below).

Diagrams and tables are fine, while Figure 1 needs several ameliorations (see below).

Comments and suggestions are listed hereinafter.

TEXT

L111: Here, a short geological-geomorphological outline should be added in the section as this could help not Asian readers to understand the main outcrops and active/inactive processes along the barred Nakdong River.

L113-114: Since it is a monsoon region, indicate the climate classification sensu Köppen (1936) and consider reading and citing the following articles:

  • Köppen W (1936) Das geographische System der Klimate. In: Köppen W and Geiger R (eds) Handbuch der Klimatologie. Berlin: Gerbru¨der Borntraeger, Vol. I, Part C, 44.
  • Kottek M, Grieser J, Beck C, et al. (2006) World map of the Köppen-Geiger climate classification updated. Meteorologische Zeitschrift 15: 259-263.
  • Trewartha GT and Horn LH (1980) An Introduction to Climate. 5th ed. New York: McGraw Hill, 416.

L61-64: What is the date of weirs construction? Specify here, as it is not clear: considering figures 2, 3, and 4, the date is after 2008 and (maybe) before 2012. But, the operating date of each weir is the same or not?

L134: "... L ..." The symbol for liter is "l" (lowercase letter): check it in all the text.

FIGURES

Figure 1: The figure is a simple wireframe, but definitely unreadable (for not Korean readers) and incomplete: sea and land are indistinguishable and have the same line thickness... Is there a sea? What is the name of rivers, capes, and bays? Add the main ones among them, especially the Nakdong River (!). Have all the river weirs and dams a name? I guess yes, then add all of them to the map: in total ten barrages, that is two dams and eight weirs. Place the geographic coordinates (labeling of latitude and longitude) along the frame edges or at two opposite vertices. Insert the two items of the legend into a solid small frame, as the line of the territory is truncated. Add letters (A) and (B) - as in figs 2, 3, and 4 - in these two figures, the smaller and the larger, respectively. In the caption, specify what is the geographic coordinate system (e.g., WGS84 or other).

TABLES

The two tables work well.

Author Response

  First of all, the co-authors and I would like to appreciate to reviewers and editor for their kind response and comments to improve our manuscript draft (sustainability-1612523). According to the comments, we would like to submit a written reply. Responses (corrections) for Reviewer 3 are tracking by word program.

â–ª The manuscript describes the possible longitudinal changes of zooplankton communities in a river-reservoir hybrid system, that is a particular transition environment, due to anthropogenic modification for the construction of some weirs along the Nakdong River. The monitored zooplankton community trends in the 2002-2018 period could allow a better understanding of the influences of these manmade structures on river connectivity, as declared in the two main objectives of this research.

Even if I am not an ecologist but a transition environments geomorphologist with expertise on dam effects on the landscape, however, I noted some points and a figure to improve so that help the not Asian reader to better understand the study area features.

Generally, the manuscript is quite well structured: the Introduction, Materials and Methods, Results, and Discussion sections are congruent and sufficient. The Conclusions section is very poor, only six lines (...): I suggest extending it summarizing better the main results.

References are sufficient (53) and pertinent, but a few more citations are suggested (see below).

Diagrams and tables are fine, while Figure 1 needs several ameliorations (see below).

Comments and suggestions are listed hereinafter.

==> Thank you for the comments. We would revise a manuscript following your comments.

==> We also revised conclusion part.

Long-term monitoring results performed at 6 longitudinal survey points of the river for 14 years helped to understand how the similarity of zooplankton community changes due to the construction of the lateral structures in Nakdong River. A stagnant and increased water body caused by RRHSs functions as a zooplankton habitat, but the longitudinal gradient, a unique characteristic of zooplankton in running water, had disappeared. Significant changes have also occurred in the patterns of previously thriving zooplankton taxa communities. Large-scale anthropogenic disturbances in rivers, such as 4LRPs, should be thoroughly evaluated before and after construction and managed with constant advice from experts in various fields. Nevertheless, the biodiversity of rivers will be impacted. Although minimizing disturbances should be the primary goal, zooplankton can be a useful indicator of river ecosystem health.

TEXT

â–ª L111: Here, a short geological-geomorphological outline should be added in the section as this could help not Asian readers to understand the main outcrops and active/inactive processes along the barred Nakdong River.

==> Thank you for your comments. We wrote more detail description about Nakdong River.

The Nakdong River has a gentle slope, and the elevation difference of the mainstream is less than 100 m. In the upper and middle streams, it flows along the tectonic line between the mesozoites, and in the downstream it flows through the porphyrite zone.

â–ª L113-114: Since it is a monsoon region, indicate the climate classification sensu Köppen (1936) and consider reading and citing the following articles:

Köppen W (1936) Das geographische System der Klimate. In: Köppen W and Geiger R (eds) Handbuch der Klimatologie. Berlin: Gerbru¨der Borntraeger, Vol. I, Part C, 44.

Kottek M, Grieser J, Beck C, et al. (2006) World map of the Köppen-Geiger climate classification updated. Meteorologische Zeitschrift 15: 259-263.

Trewartha GT and Horn LH (1980) An Introduction to Climate. 5th ed. New York: McGraw Hill, 416.

==> We checked the data from Kottek et al., 2006 to reflect the latest climatic zones among the data you recommended. According to this, it is not possible to agree that the Nakdong River basin belongs to Cwa (warm temperature, dessert, and hot summer). This is because the average annual rainfall for the watershed is 1,200 mm, of which more than 60% falls within three months. Therefore, we would maintain the same description of the Nakdong River as described in other existing papers.

â–ª L61-64: What is the date of weirs construction? Specify here, as it is not clear: considering figures 2, 3, and 4, the date is after 2008 and (maybe) before 2012. But, the operating date of each weir is the same or not?

==> According to the literatures, the groundbreaking ceremony for 4LRPs was held on December 29, 2008. After that, it was confirmed that the construction of the dam was completed, and the dam was opened around November 2011. However, since the data provided by the institute that manages them is from August 2012, the exact start date of operation could not be confirmed. Therefore, we did not use data from 2009 to 2011.

â–ª L134: "... L ..." The symbol for liter is "l" (lowercase letter): check it in all the text.

==> Thank you for your comments. However, the unit L, which can be written with l, was adopted by the CIPM (International Committee for Weights and Measures) in 1979 to avoid l being confused with the number 1. And recent papers also used L instead of l (lower case ‘L’). An example of such kind of use in relation to unit (litter) might be found in: https://doi.org/10.1016/j.watres.2022.118267.

FIGURES

â–ª Figure 1: The figure is a simple wireframe, but definitely unreadable (for not Korean readers) and incomplete: sea and land are indistinguishable and have the same line thickness... Is there a sea? What is the name of rivers, capes, and bays? Add the main ones among them, especially the Nakdong River (!). Have all the river weirs and dams a name? I guess yes, then add all of them to the map: in total ten barrages, that is two dams and eight weirs. Place the geographic coordinates (labeling of latitude and longitude) along the frame edges or at two opposite vertices. Insert the two items of the legend into a solid small frame, as the line of the territory is truncated. Add letters (A) and (B) - as in figs 2, 3, and 4 - in these two figures, the smaller and the larger, respectively. In the caption, specify what is the geographic coordinate system (e.g., WGS84 or other).

==>We revised a Figure 1. Although we would like to add a names like weirs, dams, an estuary bank, and etc, we only revised part of them. Because the map would be complicated. And coordinate system also was added in the caption.

Figure 1. Description of study sites in the Nakdong River. The yellow dots are the study sites, and the red rectangles are the constructed weirs from 4 large river projects (4LRPs) (coordinate system: PCS ITRF2000 TM).

â–ª TABLES

The two tables work well.

Round 2

Reviewer 1 Report

Reviewer comments

I would like to thank the authors for the review; the manuscript has been greatly improved. However, there are some details that need to be addressed.

Line 22-24: “Our study demonstrates that…”. This sentence may be changed to something like: “Our results demonstrate the negative effect of lateral infrastructures on zooplankton populations due to river fragmentation and habitat alteration”.

Line 38: The cohesion of this paragraph could be improved. Sentences jump from one topic to the next without a clear structure. Maybe you can start talking about the effect of barriers in habitat, then fish and finish with zooplankton, linking it to the next paragraph, in which you only focus on zooplankton.

Line 40: Change “biological effects” by “studies on biological effects”.

Line 41: To improve the structure, this line can be changed to something like: “In fishes, decreasing populations and higher proportion of exotic species were reported”.

Line 43: “fragmentation” instead of “Fragmentation”.

Line 78-81: In this sentence, you explain what you are doing so it may be better place at the end of the introduction, where you explain your objective and methods. Here the sentence is isolated and confusing, as you have not defined your aim yet.

Line 104: “rivers” instead of “river".

Line 105: change “4” with “four” and add “built”: “four large river projects (4LRPs) built from…”.

Line 224: “As a result” instead of “As a results”.

Table 2 caption: Add “of the” and change word order: “after seasonal decomposition of the zooplankton community dataset…”.

Line 467-469: Change the word order: “Ecosystem changes such as 4LRPs…”.

Line 469: Change “rotifers were increased” for “rotifers increased”.

Author Response

Reviewer 1

â–ª I would like to thank the authors for the review; the manuscript has been greatly improved. However, there are some details that need to be addressed.

  • Thank you for your positive comments on our manuscript titled ‘Impact of river-reservoir hybrid system on zooplankton community and river connectivity’ for publication in Sustainability. We did our best to revise manuscript in accordance with your comments and hope that the polished manuscript will be useful to a wide range of readers.

â–ª Line 22-24: “Our study demonstrates that…”. This sentence may be changed to something like: “Our results demonstrate the negative effect of lateral infrastructures on zooplankton populations due to river fragmentation and habitat alteration”.

  • We replaced the word.

â–ª Line 38: The cohesion of this paragraph could be improved. Sentences jump from one topic to the next without a clear structure. Maybe you can start talking about the effect of barriers in habitat, then fish and finish with zooplankton, linking it to the next paragraph, in which you only focus on zooplankton.

  • Thank you for your comments. We added two sentences to complement the connectivity after the first sentence of the second paragraph of the introduction.
  • Weir and dam construction cause species isolation [8]. Habitat transformation caused by a reduction in river connectivity favor generalist species over more habitat specific species [9].

[8]   Falke, J.A.; Gido, K.B. Spatial effects of reservoirs on stream fish assemblages in the Great Plains, USA. River Res. Appl. 2006, 22, 55-68.

[9]   Branco, P.; Segurado, P.; Santos, J.M.; Pinheiro, P.; Ferreira, M.T. Does longitudinal connectivity loss affect the distribution of freshwater fish? Ecol. Eng. 2012, 48, 70-78.

â–ª Line 40: Change “biological effects” by “studies on biological effects”.

  •  We replaced it.

â–ª Line 41: To improve the structure, this line can be changed to something like: “In fishes, decreasing populations and higher proportion of exotic species were reported”.

  • We replaced the sentence.

â–ª Line 43: “fragmentation” instead of “Fragmentation”.

  • We replaced the word.

â–ª Line 78-81: In this sentence, you explain what you are doing so it may be better place at the end of the introduction, where you explain your objective and methods. Here the sentence is isolated and confusing, as you have not defined your aim yet.

  • Thank you for your comments. Your comments make sense, but we would like to maintain the sentence. Because the sentences were written based on the references related to the Nakdong River and 4LRPs.

â–ª Line 104: “rivers” instead of “river".

  • We replaced the word.

â–ª Line 105: change “4” with “four” and add “built”: “four large river projects (4LRPs) built from…”.

  • We replaced the words.

â–ª Line 224: “As a result” instead of “As a results”.

  • We replaced the word.

â–ª Table 2 caption: Add “of the” and change word order: “after seasonal decomposition of the zooplankton community dataset…”.

  • We replaced the word.

â–ª Line 467-469: Change the word order: “Ecosystem changes such as 4LRPs…”.

  • We changed it according to your comment.

â–ª Line 469: Change “rotifers were increased” for “rotifers increased”.

  • We changed it.

Reviewer 2 Report

All requested changes were addressed.

Author Response

Thank you for your positive comments on our manuscript titled ‘Impact of river-reservoir hybrid system on zooplankton community and river connectivity’ for publication in Sustainability.  We did our best to revise manuscript in accordance with your comments and hope that the polished manuscript will be useful to a wide range of readers.